# Clinical Characteristics of Anti-TIF-1γ Antibody-Positive Dermatomyositis Associated with Malignancy

**DOI:** 10.3390/jcm11071925

**Published:** 2022-03-30

**Authors:** Yumi Harada, Masaki Tominaga, Eriko Iitoh, Shinjiro Kaieda, Takuma Koga, Kiminori Fujimoto, Tomonori Chikasue, Hitoshi Obara, Tatsuyuki Kakuma, Hiroaki Ida, Tomotaka Kawayama, Tomoaki Hoshino

**Affiliations:** 1Division of Respirology, Department of Medicine, Neurology and Rheumatology, Kurume University School of Medicine, Kurume 830-0011, Japan; yumi_yoshida@med.kurume-u.ac.jp (Y.H.); eor1i2i2@icloud.com (E.I.); kaieda@med.kurume-u.ac.jp (S.K.); koga_takuma@med.kurume-u.ac.jp (T.K.); ida@med.kurume-u.ac.jp (H.I.); kawayama_tomotaka@med.kurume-u.ac.jp (T.K.); hoshino@med.kurume-u.ac.jp (T.H.); 2Department of Radiology, Center for Diagnostic Imaging, Kurume University School of Medicine, Kurume 830-0011, Japan; kimichan@med.kurume-u.ac.jp (K.F.); chikasue_tomonori@med.kurume-u.ac.jp (T.C.); 3Biostatistics Center, Kurume University, Kurume 830-0011, Japan; obara_hitoshi@kurume-u.ac.jp (H.O.); tkakuma@med.kurume-u.ac.jp (T.K.)

**Keywords:** dermatomyositis (DM), anti-transcriptional intermediary factor 1 (TIF-1γ) antibody, anti-aminoacyl tRNA synthetase (ARS) antibody, anti-melanoma differentiation-associated gene 5 (MDA-5) antibody, skin manifestation

## Abstract

We retrospectively analyzed the clinical and laboratory data of patients diagnosed with anti-transcriptional intermediary factor 1 (TIF-1γ) antibody-positive polymyositis (PM)/dermatomyositis (DM) to clarify the characteristics of this disease. We identified 14 patients with TIF-1γ antibody-positive DM (TIF-1γ DM), 47 with anti-aminoacyl-tRNA synthetase antibody (ARS)-positive PM/DM, and 24 with anti-melanoma differentiation-associated gene 5 antibody (MDA-5)-positive PM/DM treated at the Kurume University Hospital between 2002 and 2020. Patients with TIF-1γ DM were significantly older than the other two groups. Nine patients with TIF-1γ DM were female, thirteen patients had DM, and one had clinically amyopathic DM. Primary malignant lesions were lung (3), uterus (2), colon (2), breast (2), ovary (1), lymphoma (1), and unknown (2). Cutaneous manifestation and dysphagia were the most common symptoms in TIF-1γ DM. Erythema (9/14), the V-neck sign (8/14), heliotrope (9/14), and nailfold telangiectasia (14/14) were significantly more common in TIF-1γ DM. Furthermore, no patients with TIF-1γ DM had interstitial lung abnormality on high-resolution CT. In patients with TIF-1γ DM, the frequency of dysphagia and unusual erythema, particularly that which spreads from the trunk, and nailfold telangiectasia, were characteristic findings. In most patients with TIF-1γ DM, it is necessary to administer other immunosuppressive drugs along with glucocorticoids.

## 1. Introduction

Although dermatomyositis (DM) has been recognized as an autoimmune disease, several novel specific autoantibodies have been discovered recently. These include anti-aminoacyl tRNA synthetase (ARS) antibodies, such as the anti-Jo-1 and anti-Mi-2 antibodies, anti-melanoma differentiation-associated gene 5 (MDA-5) antibody, anti-transcriptional intermediary factor 1 (TIF-1γ) antibody, anti-nuclear matrix protein 2 antibodies, and anti-small ubiquitin-like modifier-1 activating enzyme antibody [1,2]. Recently, inflammatory myopathy is classified based on these myositis-specific autoantibodies because each group has unique characteristics [3], and anti-synthetase syndrome is a new concept to be a differentiated nosological disease from DM. [4] For example, patients with MDA-5 antibody-positive DM (MDA-5-DM) often present with clinically amyopathic DM (CADM), which is frequently complicated by rapidly progressive interstitial lung disease (ILD) and have a bad prognosis [5]. Alternatively, anti-TIF-1γ antibody-positive DM is closely associated with cancer-associated DM, and patients present with skin rashes, proximal muscle weakness, and dysphagia [6].

In this study, we attempted to identify the clinical characteristics of anti-TIF-1γ-associated DM. We encountered 14 cases of anti-TIF-1γ-positive DM (TIF-1γ DM), and herein, we present the clinical characteristics of these patients. These results may assist physicians in treating and determining the salient clinical checkpoints.

## 2. Materials and Methods

This study included 85 consecutive patients diagnosed with PM/DM between 2002 and 2020 at the Kurume University Hospital. Clinical data, cumulative disease manifestations, laboratory investigations, associated diseases, therapy, clinical course, disease complications, and outcomes were retrospectively recorded from case notes.

Forty-seven patients tested positive for the anti-ARS antibody; 24 tested positive for the MDA-5 antibody; and 14 tested positive for TIF-1γ antibodies. The diagnosis of PM or DM was confirmed according to the Bohan and Peter criteria [7]. The diagnosis of CADM was based on the presence of a skin rash characteristic of DM and no clinical evidence of a muscular disorder or myositis. After the diagnosis of PM/DM, we attempted to detect the malignant lesion using serum tumor markers, enhanced CT scans, and gastrointestinal fiber scope examinations.

We collected initial clinical, physiological, and radiological data for all patients. Patients with missing data were excluded. All clinical (physical examinations), serological, and demographic data were collected retrospectively from the medical records at the first visit. 

This single-center study was conducted in accordance with the tenets of the Declaration of Helsinki and involved a retrospective review of clinical records. The protocol was approved by the ethics committee of the Kurume University (approval no. 19003; 15 April 2017). The requirement for patient approval or informed consent was waived owing to the retrospective nature of the study.

### 2.1. Blood Tests

ELISA kits were used to measure anti-ARS antibody (cut-off value = 25; MESACUP anti-ARS test, MBL, Nagoya, Japan), anti-MDA-5 antibody (cut-off value = 32; MESACUP anti-ARS test, MBL), and anti-TIF-1γ antibody (cut-off value = 60; MESACUP anti-TIF-1γ test, MBL) levels.

### 2.2. Evaluation of High-Resolution Computed Tomography (HRCT) Findings and Patterns

HRCT images of the patients were obtained at end-inspiration using various scanners with the patient in the supine position. The protocols included 0.5–1.25 mm collimation sections reconstructed with a high-spatial-frequency algorithm at 1 or 2 cm intervals. The images were photographed at window settings appropriate for viewing the lung parenchyma (window level from −650 to −700 Hounsfield units [HU]; window width from 1200 to 1500 HU) and mediastinum (window level from 400 to 500 HU; window width from 20 to 40 HU).

Patients’ chest HRCT images were retrospectively evaluated by two expert chest radiologists (KF and TC, 34 and 6 years of experience in the diagnosis of ILD, respectively). The radiologists were aware of the patients’ ARS or melanoma differentiation-associated gene 5 antibody-associated interstitial lung disease (MDA-ILD) diagnoses but were blinded to other clinical findings and outcomes.

In this study, two radiologists assessed the presence or absence of interstitial lung abnormality (ILA) on HRCT.

### 2.3. Statistical Analysis

A simple regression model was employed to determine the associations between clinical characteristics, initial treatments, HRCT findings, and serological data and each DM antibody. All values are expressed as the mean ± standard deviation. Group comparisons were made using Fisher’s exact tests, Statistical significance was set at *p* < 0.05. All data were analyzed using JMP^®^ Pro 13.2.0 (SAS Institute Inc., Cary, NC, USA).

## 3. Results

### 3.1. Clinical Characteristics

The clinical characteristics of patients with ARS, MDA-5, and anti-TIF-1γ antibody-positive DM are summarized in Table 1. The mean age of the TIF-1γ group was 68.6 ± 10.7 years, and nine patients were female. The mean age of the TIF-1γ group was significantly higher than that of the ARS-DM and MDA-5-DM (TIF-1γ-negative DM) groups. Thirteen patients had DM, and one had CADM. This proportion of DM was significantly higher than that of TIF-1γ-negative DM. Cancer-associated myositis is defined malignancy develops within a year or two of a diagnosis of myositis. ARS-DM, MDA5-DM and TIF1γ-DM patients had malignancy 5, 0 and 12, respectively. In ARS-DM patients, the primary lesions were located in the uterus (*n* = 1), prostate (*n* = 1), and breast (*n* = 3). In TIF1γ-DM patients, the primary lesions were located in the lung (*n* = 3), uterus (*n* = 2), stomach (*n* = 1), colon (*n* = 2), breast (*n* = 2), and ovary (*n* = 1), or was a lymphoma (*n* = 1); malignant lesions were not detected in two patients. Interestingly, two patients were diagnosed with cancer-related DM after starting chemotherapy for primary cancer. The mean duration from onset to first visit was 119.36 ± 155.87 days, and the mean duration from the initial treatment visit was 121.5 ± 437.31 days. The ‘onset’ was defined as the time when the patient recognized any manifestations, such as dysphagia or cutaneous manifestations by patient’s home diary. The definition of ‘first visit’ was the visit to any clinic or hospital. Therefore, some medical services were not provided at our hospital. Three patients were diagnosed with malignancy at the time of DM diagnosis. 

The clinical findings of patients with TIF-1γ DM are summarized in Table 2. Dyspnea on effort was not noted in patients with TIF-1γ DM but was noted in TIF-1γ-negative patients, demonstrating a significant difference (*p* < 0.005). Cutaneous manifestations (100%) and dysphagia (74%) were more frequently observed in TIF-1γ DM (both *p*-values were <0.001). During physical examinations, neither Raynaud’s phenomenon nor arthritis/arthralgia were observed in patients with TIF-1γ DM. The V-neck sign (57%, *p* < 0.001), erythema (64%, *p* < 0.001), heliotrope (64%, *p* = 0.020), and nailfold bleeding (*p* < 0.001) were frequently noted. Skin manifestations in patients with TIF-1γ DM were unusual compared with TIF1γnegative DM patients. Especially, characteristics of skin manifestation of TIF-1γ DM patients were widespread, more dark-red in color, and scattered throughout the body. They may manifest as an erythematous-violaceous rash (Figure 1a,b) or a crusted erosive lesion (Figure 1c). The V-neck sign is shown in Figure 1d, and erythema was more severe compared to that in patients with TIF-1γ-negative DM, its characteristics were spreading on the trunk and bilateral arms, and occasionally accompanied by dark pigmentation (Figure 1e).

### 3.2. Laboratory, Pulmonary Function Test, and Computed Tomography (CT) Findings

The laboratory findings are shown in Table 2. The mean ferritin level in patients with TIF-1γ DM was 3875 ± 2646 IU/L, which was higher than that in patients with ARS-DM and MDA-5-DM. The ferritin level probably depends on the malignancy potential; therefore, the mean ferritin level in patients with TIF-1γ DM was significantly higher than that in patients with TIF-1γ-negative DM (*p* < 0.001). The mean Krebs von den Lungen 6 (KL-6) level was 1502.8 ± 2075.8 U/mL, and this was lower than that in patients with ARS- and MDA-5-positive DM. None of the patients had ILA (*p* < 0.001), and the low KL-6 level might account for this result.

### 3.3. Treatment

Table 3 shows the disease behavior, treatment, primary cancer, and clinical course. All patients, except for two, were treated with prednisolone (PSL). Two patients did not want to receive aggressive therapy for primary cancer; therefore, they were administered conservative care (cases 3 and 8). Two patients were administered PSL with tacrolimus (TAC) (cases 4 and 7), and one was administered azathioprine (AZP) (case 14). Unfortunately, four out of nine patients did not improve with PSL only (cases 2, 6, 11, and 12) and required more aggressive therapy. Thus, TAC was added for three patients (cases 6, 11, and 12), and high-dose intravenous immunoglobulin (IVIG) was added for one patient (case 2). One of the four patients who were treated with PSL + TAC after single PSL therapy (case 6) did not show improvement. AZP was added instead of TAC, but DM activity could not be controlled. After treatment with IVIG, the patient’s skin manifestations improved. Three of the fourteen patients were of advanced age and did not want to receive anti-cancer therapy (cases 3, 5, and 7). After anti-cancer therapy, three patients showed complete remission (cases 2, 6, and 7) and two showed partial remission (cases 13 and 14), but five patients showed deterioration (cases 1, 4, 10, 11, and 12). In the patients undergoing aggressive chemotherapy for cancer, disease activity was controlled in four out of five patients and skin manifestations and dysphagia were improved (cases 2, 6, 13, and 14). In contrast, skin manifestations deteriorated in two out of five patients in whom disease activity was not controlled (cases 4 and 9). Although disease activity was not controlled, in three out of five patients, skin manifestations were controlled by PSL and/or other immunosuppressants (cases 1 and 10). Many cases moved to another hospital, so we could not trace their skin lesion and outcome. However, to our knowledge, although five cases showed deterioration of primary cancer, eight cases were controlled their skin lesion. 

Figure 2 shows the follow-up data of CK level. All of the cases decreased the CK level after treatment.

Case 13 showed the typical clinical course of TIF-1γ DM; the patient complained of muscle weakness, skin eruption, and dysphagia. (Figure 3) Lung nodules with lymphadenopathy were detected and diagnosed as small cell lung cancer. He was immediately treated with aggressive chemotherapy (carboplatin+ etoposide), but his symptoms temporally deteriorated. Consequently, PSL was started. His symptoms gradually improved with the effect of chemotherapy and PSL. Interestingly, primary lung cancer lesion and lymphadenopathy decreased in size, and the titer of TIF-1γ antibody also decreased after 3 months. 

## 4. Discussion

We presented 14 TIF-1γ DM patients and presented the characteristics of this disease. In most of the cases, we detected malignancy at the same time of disease onset, but in some cases, we did not. Characteristics of TIF-1γ DM patients were (1) never complicated with ILD, (2) cutaneous lesion was severe than other TIF-1γ-negative DM, (3) often complicated with dysphagia. Cutaneous lesions showed spreading wide range, dark-colored and dirty. Additionally, they showed resistance to any treatments. 

The TIF-1γ antibody was first reported as a member of the *TIF-1* gene family in 1999 [8]. Since then, anti-TIF-1 antibodies have been reported as anti-155/140 and anti-p155 antibodies that target a 155-kDa nuclear proteins with or without a 140-kDa protein [2,8]. The 155-kDa and 140-kDa proteins were subsequently identified as TIF-1γ and TIF-1α, respectively [9,10,11]. TIF-1γ plays a role in transcriptional elongation, DNA repair, cell differentiation, embryonic development, and mitosis. Moreover, TIF-1γ may suppress various cancers via the TGF-β/Smad and Wnt/β-Catenin signaling pathways [12].

Hill et al. confirmed a markedly increased risk of malignancy in patients with DM compared with the general population, with 32% of patients with DM and 15% of patients with patients with PM had malignancies in that study [13]. 

The association between malignancy and DM is well-described in the literature [14,15,16,17]. One published series of patients with DM reported an incidence of cancer ranging from 9% to 42% [15,16]. The most frequent are ovarian, breast, lung, gastric, and colorectal tumors, as well as lymphomas in dermatomyositis, lung and urinary bladder cancers, and lymphomas in polymyositis [18]. Most cancers preceded, occurred concurrently, or occurred after the onset of DM, but is usually recognized within 3 years of DM diagnosis. However, most of the cases may be diagnosed within 12 months [18]. In our study, cancer was recognized in one case after surgery for the primary lesion, and one patient was under anti-cancer therapy. In general, clinical risk factors for malignancy in DM include older age, progressive disease, longer duration of symptoms before diagnosis, initiation of therapy after 24 months of muscle weakness, cardiac issues, pulmonary problems, male sex, severe skin manifestations, dysphagia, resistance to treatment, and a history of malignancy with risk of relapse and absence of ILD [14,18].

In this paper, we have discussed the differentiation of cutaneous lesions in patients with TIF-1γ-positive from those in patients with other forms of DM. In general, typical skin lesions such as the Gottron sign, centrofacial erythema, and typical erythema on the upper arms and forearms were common in patients with DM and malignancy. The Gottron papules were distinctly hyperkeratotic and scaly [17,18,19]. Daly et al. reported that facial dermatomyositis and the shawl sign were significantly more common in patients with TIF-1γ DM [20]. In our patients, the V-neck sign (shawl sign) was present in 8 out of 14 patients (54%), and this sign was more commonly seen than other symptoms. Ulcerated and eroded skin lesions were common in MDA-5-DM, but not in older patients with TIF-1γ DM [21]. Fiorentino et al. reported that characteristic cutaneous features have been reported, including psoriasiform lesions or so called ‘red-on-white’ patches (hypopigmented macules/patches associated with focal, often follicular, telangiectatic erythema) [22]. Our patients’ cutaneous manifestations were similar to those in the reports [17,18,19,20,21,22], but these findings were severe compared with those in patients with TIF-1γ-negative DM. For example, skin lesions tended to be widespread, more dark-red in color, and scattered throughout the body. 

DM with malignancy sometimes shows severe and rapidly progressive muscle involvement and distal muscle weakness. Involvement of the oropharyngeal musculature, especially dysphagia, occurs more frequently, and respiratory impairment caused by respiratory muscle weakness has been shown to be a characteristic neurological feature [18,23]. In our study, 8 out of 14 patients showed dysphagia, and this high incidence was similar to that in previous reports [18,23]. 

Although ILD is a crucial complication in patients with DM, TIF-1γ DM is generally not complicated by ILD [24]. In fact, there were no cases of ILD involvement in our study. This was most likely because the KL-6 level was not high in all cases. These data are consistent with those of previous reports [25]. 

In terms of treatment, it has been recognized that the treatment for malignancies should be prioritized [26]. Corticosteroids are effective to TIF-1γ DM, but some cases show steroid resistance. In these cases, a variety of non-steroid drugs, including methotrexate, antimalarial drugs, mycophenolate, AZP, and CyA, might be effective. The efficacy of immunoglobulins, rituximab, and tumor necrosis factor-α inhibitors has been described for patients with multiple drug resistance [27,28]. In our study, patients whose cancer was controlled showed a relatively good response in terms of dysphagia and cutaneous manifestations. However, many patients required immunosuppressive therapy to control disease activity, especially those with cutaneous lesions and dysphagia. One patient showed multiple drug resistance, but IVIG was effective. We considered IVIG treatment when patients with TIF-1γ DM showed drug resistance. 

This study had some limitations. This was a single-center, retrospective study with a small sample size, and the study design could have introduced bias and obscured statistical significance. A larger population should be examined in future studies using a prospective research model.

## 5. Conclusions

DM is known to be associated with malignancy; thus, cancer screening is necessary for patients with DM. The characteristics of skin manifestation of TIF-1γ DM patients were widespread, more dark-red in color, and scattered throughout the body. When the skin manifestation is severe, is complicated by dysphagia in the absence of ILA, and shows resistance to corticosteroid therapy, clinicians should check the TIF-1γ antibody levels and perform cancer screening.

## Figures and Tables

**Figure 1 jcm-11-01925-f001:**
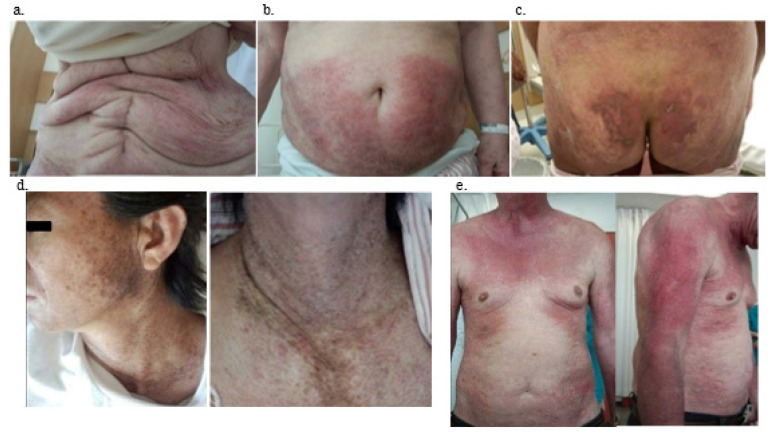
Representative skin manifestations of anti-TIF1γ positive DM patients. (**a**) Case 11, (**b**) case 4, (**c**) case 2, (**d**) case 6, (**e**) case 5. TIF-1γ, anti-transcriptional intermediary factor 1; DM, dermatomyositis; (**a**,**b**): Extensive erythema over the anterior torso. (**c**): Widespread, crusted erosive lesions on the buttock. (**d**): Photo-distributed confluent dark red-purple erythema on the upper chest (V-neck sign). (**e**): Erythematous-violaceous rash on sun-exposed areas on the upper part of the trunk and proximal extensor surfaces of the upper limbs.

**Figure 2 jcm-11-01925-f002:**
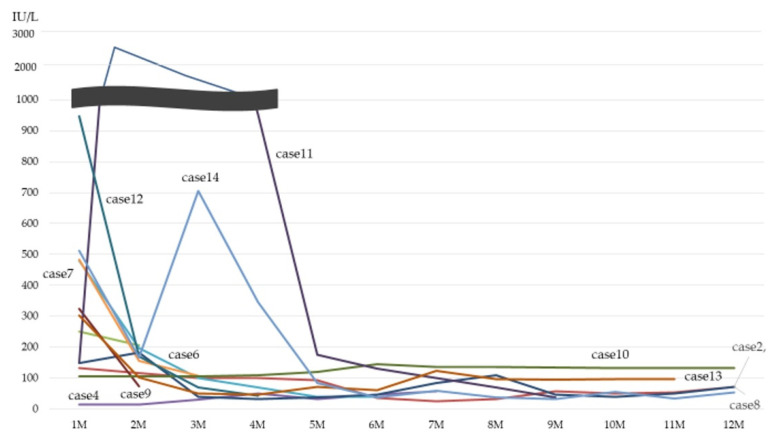
Follow-up data of CK level. CK: creatinine kinase.

**Figure 3 jcm-11-01925-f003:**
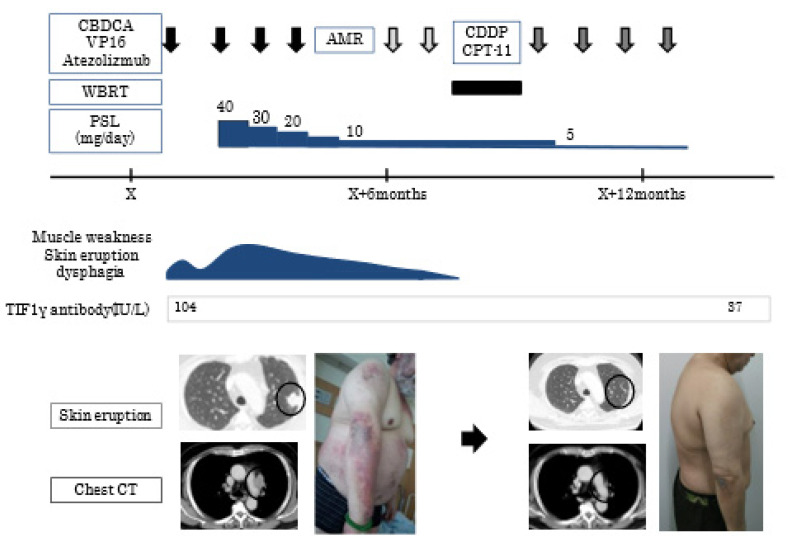
Clinical course of case 13. Chest CT shows a lung nodule(circle) in the left upper lobe with lymphadenopathy(arrow). CBDCA, carboplatin; VP-16, etoposide; AMR, amrubicin; CDDP, cisplatin; CPT-11, irinotecan; WBRT, whole brain radiotherapy; PSL, prednisolone; TIF-1γ, anti-transcriptional intermediary factor 1; CT, computed tomography.

**Table 1 jcm-11-01925-t001:** Clinical characteristics of patients with anti-ARS, anti-MDA-5, and anti-TIF-1γ-positive DM at onset.

	ARS (47)	MDA-5 (24)	TIF-1γ (14)	*p*-Value
Patient characteristics				
Age (years)	59.7 ± 10.2	53.9 ± 11.9	68.6 ± 10.7	0.001 *
PM/DM/CADM	9/29/9	0/13/11	0/13/1	0.001 *
Sex (male/female)	10/37	8/16	5/9	0.435
Duration from onset to first visit (days)	226.6 ± 422.18	92 ± 182.06	119.36 ± 155.87	0.535
Duration from first visit to treatment initiation (days)	131.45 ± 609.84	9.08 ± 10.75	121.5 ± 437.31	0.736
Outcome (death/alive)	45/2	14/10	12/2	0.812
Malignacy	5 (11%)	0 (0%)	12 (86%)	<0.001 *
Clinical Symptoms at the onset				
Dyspnea on effort	26 (55%)	13 (52%)	0 (0%)	<0.001 *
Fever	14 (30%)	13 (52%)	2 (14%)	0.068
Myalgia	19 (40%)	11 (46%)	3 (21%)	0.130
Skin manifestation	10 (21%)	14 (58%)	14 (100%)	<0.001
Dysphagia	0 (0%)	0 (0%)	8 (71%)	<0.001 *

Asterisks indicate statistical significance (*p* < 0.05). PM, polymyositis; DM, dermatomyositis: CADM, clinically amyopathic dermatomyositis; ARS, autoantibodies against aminoacyl-tRNA synthetases; MDA-5, anti-melanoma differentiation-associated gene 5 antibody; TIF-1γ, anti-transcriptional intermediary factor 1.

**Table 2 jcm-11-01925-t002:** Physical examination and laboratory data of patients with anti-ARS, anti-MDA-5, and anti-TIF-1γ-positive DM at onset.

	**ARS (47)**	**MDA-5 (24)**	**TIF-1γ (14)**	***p*-Value**
Physical examination				
V-neck sign (shawl sign)	1 (2%)	5 (21%)	8 (57%)	<0.001 *
Raynaud phenomenon	7 (15%)	2 (8%)	0 (0%)	0.063
Gottron’s sign	20 (43%)	24 (100%)	6 (43%)	0.223
Reverse Gottron’s sign	2 (4%)	17 (71%)	21(3%)	0.752
Arthralgia	20 (43%)	16 (67%)	0 (0%)	0.080
Erythema	3 (6%)	2 (8%)	9 (64%)	<0.001 *
Mechanics hand	21 (45%)	15 (62%)	4 (29%)	0.083
Heliotrope	3 (6%)	20 (83%)	9 (64%)	0.020
Nailfold bleeding	12 (26%)	16 (67%)	14 (100%)	<0.001 *
Muscle weakness	16 (34%)	18 (75%)	9 (64%)	0.301
Serological examination				
CRP (mg/dL)	2.37 ± 5.63	0.64 ± 0.84	0.5 ± 0.6	0.176
LDH (IU/L)	375.0 ± 107.4	393.0 ± 134.0	362 ± 108	0.476
CK (mg/dL)	45 ± 485	338 ± 540	754 ± 822	0.804
KL-6 (U/mL)	1327.2 ± 493.1	686.0 ± 489.6	209 ±56	0.110
Ferritin (ng/mL)	306.7 ± 114.9	938.9 ± 915	3875 ± 2646	<0.001 *
Radiological findings				
ILA on HRCT	47 (100%)	24 (100%)	0 (0%)	<0.001 *

Asterisks indicate statistical significance (*p* < 0.05). PM, polymyositis; DM, dermatomyositis; CADM, clinically amyopathic dermatomyositis; ARS, autoantibodies against aminoacyl-tRNA synthetases; MDA-5, anti-melanoma differentiation-associated gene 5 antibody; TIF-1γ, anti-transcriptional intermediary factor 1; ILA, interstitial lung abnormality; HRCT, high-resolution computed tomography; CRP, c-reactive protein; LDH, lactate dehydrogenase; CK, creatine phosphokinase; KL-6, Krebs von den Lungen 6.

**Table 3 jcm-11-01925-t003:** Summary of patients with TIF-1γ-positive DM.

	Sex	Age	Skin Lesion	Cancer Lesion	Treatment	Primary Lesion	Overall Outcome
1	F	71	improved	PD	PSL	uterus	worse
2	F	54	improved	CR	PSL→PSL + IVIG	uterus	worse
3	M	62	unknown	no treatment	none	unknown	unknown
4	F	74	deteriorated	PD	PSL + Tac	lung	death
5	M	79	unknown	no treatment	PSL	stomach	death
6	F	60	improved	CR	PSL→PSL + Tac→PSL + AZP→PSL + IVIG	breast	unknown
7	F	64	deteriorated	no treatment	PSL + Tac	unknown	unknown
8	F	87	improved	unknown	PSL	ovary	unknown
9	M	54	deteriorated	CR	PSL	colon	unknown
10	F	70	improved	PD	unknown	breast	better
11	F	88	deteriorated	PD	PSL→PSL + Tac	lung	death
12	F	70	improved	PD	PSL→PSL + Tac	colon	unchanged
13	M	61	improved	PR	PSL	lung	better
14	M	67	improved	PR	PSL + AZP	lymphoma	better

TIF-1γ, anti-transcriptional intermediary factor 1; DM, dermatomyositis; PD, progressive disease; CR, complete remission; PR, partial remission; PSL, prednisolone; IVIG, intravenous immunoglobulin; Tac, tacrolimus; AZP, azathioprine.

## Data Availability

Not applicable.

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
