# Peer review of "Clinical Characteristics of Anti-TIF-1γ Antibody-Positive Dermatomyositis Associated with Malignancy"

_jcm, 2022, doi:10.3390/jcm11071925_

Round 1
Reviewer 1 Report
I think it’s an appropriate and meaningful article, it will help clinician manage DM/PM patients. I comment just a few things. First you mentioned anti-TIF-1γ antibody was highly associated with malignancy, but there was no data about malignancy prevalence of other type PM/DM. I suggest you add the data of malignancy prevalence of all type of PM/DM in table 1 or 2. Second, it is minor but important problem. You wrote number and percentage in wrong order in table 1(clinical symptoms) and table 2(Reverse Gottron’s sign). If you correct these points, the paper will be improved.
Author Response
Thank you for your important comments. I have added the prevalence of malignancy in Table 1. Regarding the second question, I have revised the data as per your suggestions.
Reviewer 2 Report
The manuscript is quite interesting and useful for the clinical assessment of DM patients.
Major concerns regard the style, incorrect referencing, syllable synonyms or division, and many typographical mistakes.
For example:
Abstract lacks malignancy information present in the title,
Title and Line 32: DM and PM are two different autoimmune diseases easily differentiated by clinical and myopathological features. This sentence and Title are both in contrast with the mat. & meth. and results of your study. Please modify.
Lines 33-34: Please consider recent literature considering anti-synthetase syndrome a differentiated nosological disease from both DM and PM. https://www.ncbi.nlm.nih.gov/pmc/articles/PMC8881476/ and other
Lines 54-55: please revise the number of positive patients, their sum is 86. Had one patient a double immunopositivity? Which one>?
Line 71 and Table 1: is it possible to differentiate the different ARS and show their consistency with MESACUP anti-ARS test [MBL]?
Line 105, please add ‘anti-TIF-1gamma
Lines 105-106, this sentence is unclear, please, clarify the terms of comparison and introduce the concept of cancer-associated myositis (also after line 44). The sum of cancers is 9, cancer-free=2, there is a lack of 3 patients. Please clarify. Explain how it is possible that primary cancer was diagnosed and the site of the lesion was unknown (Table 3). How many ARS and MDA-5 patients had cancer within 3 years the onset? Please add this data in Table 1-.
Lines 109-112, is the onset of which symptoms? Which was the most complained onset? Generally, it is difficult to recognize the day of disease onset. How is it calculated? Home Diary reported?
Table 1 Clinical Symptoms at the onset or first visit or other? Please, add this information to the title
Lines 97-99, probably this sentence is the journal request, please erase.
Line 122 ‘The clinical findings of patients with TIF-1γ DM are summarized in Table 2’. Please postpone the sentence after line 125
Line 129 please clarify this ‘Skin manifestations are very common in patients with 129 TIF-1γ DM, but the features are unusual.’
Fig. 1: please combine the two photograms in d., as in e. please add in the figure legend ‘of TIF1gamma DM patients’
Fig. 2 legend; CDDP compares two times, please erase. ‘with lymphadenopathy(circle)’ it is hard to see the black circle on black background, please use two white arrows
Line 205 erase with patients, please rephrase 203-205
Lines 213-215: please clarify the meaning otherwise delete
Line 221 only anti MDA5 DM is included in the comparison. Why have you used plural?
224-225, Daly is 20. Please add ‘et al’
Fiorentino is 22, not 21
Where was reference 19 cited?
233 what do you mean with ‘following’?
240-1 Ten out of 14 patients had shown (not showed), and this high incidence was similar to 240 that in previous reports. Incomplete and incorrect sentence. What sign or symptom? Where it is described in results? References?
244-5 please add information and references about KL-6.
247 Corticosteroids are effective, against what?
267, resistance instead of resistant
Table 2 Physical examination and laboratory data at the onset or first visit or other? Please, add this information to the title. The title also contains typos. How was assessed muscle weakness? Which Scale, MMT-8?
Is follow-up CK available for several patients (I acknowledge that this was a retrospective study and not all data may be available)? please add
Table 3: an overall assessment of outcomes may be feasible (better, unchanged, worse). Could you add?
Is follow-up data available (CK, MMT-8)? Please add.
The first part of the discussion should summarise and emphasize the findings of your study, rather than results from others. Suggest re-working.
You need to be more explicit about any novel findings from your study, versus what you have simply replicated from previous studies.
Author Response
→Thank you for your important comments and suggestion. I have tried to improve my manuscript based on your suggestions. Please go through the point-by-point revisions made in the manuscript.
The manuscript is quite interesting and useful for the clinical assessment of DM patients. Major concerns regard the style, incorrect referencing, syllable synonyms or division, and many typographical mistakes. For example: Abstract lacks malignancy information present in the title, →Thank you for your suggestions. I have added the information regarding malignancy in the abstract.
Title and Line 32: DM and PM are two different autoimmune diseases easily differentiated by clinical and myopathological features. This sentence and Title are both in contrast with the mat. & meth. and results of your study. Please modify. →Thank you for your suggestion. I agree with your observation., Accordingly I have deleted the word ‘PM’ from the respective areas.
Lines 33-34: Please consider recent literature considering anti-synthetase syndrome a differentiated nosological disease from both DM and PM. https://www.ncbi.nlm.nih.gov/pmc/articles/PMC8881476/ and other →Thank you for your recommendation. I have added the suggested article and the concept of ASS. (L39-41)
Lines 54-55: please revise the number of positive patients, their sum is 86. Had one patient a double immunopositivity? Which one>? →I Apologize for the misunderstanding created. I have corrected the patient number accordingly. Thank you for pointing this error in the manuscript (L52).
Line 71 and Table 1: is it possible to differentiate the different ARS and show their consistency with MESACUP anti-ARS test [MBL]? →Thank you for your question. Yes, it is possible. I have revised the information accordingly (L76.77).
Line 105, please add ‘anti-TIF-1gamma → Thank you for your suggestion. I have added ‘anti-TIF-1 gamma’ as suggested (L104).
Lines 105-106, this sentence is unclear, please, clarify the terms of comparison and introduce the concept of cancer-associated myositis (also after line 44). The sum of cancers is 9, cancer free=2, there is a lack of 3 patients. Please clarify. Explain how it is possible that primary cancer was diagnosed and the site of the lesion was unknown (Table 3). How many ARS and MDA-5 patients had cancer within 3 years the onset? Please add this data in Table 1-. →Thank you for your suggestion. I have added the definition of cancer-associated myositis and revised the number of cases. As you pointed out, two patients had a possibility to have cancer; however, we could not detect it. I have added the data regarding the cancer cases in ARS MDA-5 positive DM in Table 1 (L109-113).
Lines 109-112, is the onset of which symptoms? Which was the most complained onset? Generally, it is difficult to recognize the day of disease onset. How is it calculated? Home Diary reported? → Thank you for your question. I have presented the definition of ‘onset’ in the manuscript (L118-120). Yes, these data were dependent on the patient’s diary (L119-120).
Table 1 Clinical Symptoms at the onset or first visit or other? Please, add this information to the title → Thank you for your suggestion. The clinical symptoms were at the onset. I have changed the terms accordingly (Table1).
Lines 97-99, probably this sentence is the journal request, please erase. → Thank you for your suggestion. I have deleted the sentence.
Line 122 ‘The clinical findings of patients with TIF-1γ DM are summarized in Table 2’. Please postpone the sentence after line 125 → Thank you for your suggestion. I have revised the sentence.
Line 129 please clarify this ‘Skin manifestations are very common in patients with 129 TIF-1γ DM, but the features are unusual.’ → Thank you for your suggestion. I have added the explanation (L138-141).
Fig. 1: please combine the two photograms in d., as in e. please add in the figure legend ‘of TIF1gamma DM patients’ → Thank you for your suggestion. I have revised the figures and figure legends accordingly. (Fig 1)
Fig. 2 legend; CDDP compares two times, please erase. ‘with lymphadenopathy(circle)’ it is hard to see the black circle on black background, please use two white arrows → Thank you for your suggestion. I have deleted CDDP and changed the circle to arrows. (Fig3)
Line 205 erase with patients, please rephrase 203-205 → Thank you for your suggestion. I have revised the sentences accordingly. (Fig3)
Lines 213-215: please clarify the meaning otherwise delete → Thank you for your suggestion. I have revised the sentences. (L214-215)
Line 221 only anti MDA5 DM is included in the comparison. Why have you used plural? → Thank you for your comment. I tried to compare with not only MDA5-DM but also ARS-DM. So I used plural.
224-225, Daly is 20. Please add ‘et al’ → Thank you for pointing it out. I have added ‘et al.’ after Daly.(L282)
Fiorentino is 22, not 21 → Thank you for pointing this out. I have revised all reference numbers accordingly. I apologize for this error.
Where was reference 19 cited? → Please note that the reference 19 was cited at L-282.
233 what do you mean with ‘following’? → Thank you for pointing this out. Yes, it was unclear. I have deleted the term ‘following’.
240-1 Ten out of 14 patients had shown (not showed), and this high incidence was similar to 240 that in previous reports. Incomplete and incorrect sentence. What sign or symptom? Where it is described in results? References? →Thank you for pointing this out. I have revised the sections accordingly. Eight was correct patient number. (L298)
244-5 please add information and references about KL-6. → Thank you for your suggestion. I have added the suggested reference (ref 25).
247 Corticosteroids are effective, against what? → Thank you for pointing this out. I have added the appropriate information (L299).
267, resistance instead of resistant → Thank you for your suggestion. I have revised the term ‘resistance’ to ‘restraint’ accordingly (ref 27).
Table 2 Physical examination and laboratory data at the onset or first visit or other? Please, add this information to the title. The title also contains typos. How was assessed muscle weakness? Which Scale, MMT-8? → Thank you for your suggestions and questions. This was at onset; I have modified the appropriate sections accordingly (Table 2).
Is follow-up CK available for several patients (I acknowledge that this was a retrospective study and not all data may be available)? please add → Thank you for your suggestion. I added the follow-up data graph of CK. All of the patients showed decreased in level of CK. (Fig2)
Table 3: an overall assessment of outcomes may be feasible (better, unchanged, worse). Could you add? → Thank you for your suggestion. I have added the overall survival rates; however, most patients moved to another hospital. Therefore, I could not verify their recent condition. (204-207, Table3)
Is follow-up data available (CK, MMT-8)? Please add. → Thank you for your suggestion. I added the follow-up data graph of CK. All of the patients showed decreased in level of CK. (Fig2)
The first part of the discussion should summarise and emphasize the findings of your study, rather than results from others. Suggest re-working. → Thank you for your suggestion. I have added a short summary as the first paragraph of the discussion section. (L248-254)
Reviewer 3 Report
Dear authors
You've done some excellent work on diagnosis and treatment of DM. It is my pleasure to offer some advice for this study. I sincerely hope that it will help.
Here are some questions and suggestions:
- Your study focused on the clinical characteristics of anti-TIF-1? antibody-positive PM/DM and has described major difference between anti-TIF-1? antibody-positive and negative patients, yet the sample capacity of anti-TIF-1? antibody-positive group is the smallest of 3 groups. This may generate a statistical bias if you only study the clinical characteristics of this group.
- In your conclusion, you mentioned that clinicians should scan for cancer if patients have several manifestations. It has been known that anti-TIF-1? antibody titer is related with tumor onset, so readers may expect a more specific analysis on connections between these clinical characteristics and cancer lesion. You have only offered specific trace of one typical patient. We are expecting a more specific and statistical analysis in this part.
Author Response
1. Your study focused on the clinical characteristics of anti-TIF-1? antibody-positive PM/DM and has described major difference between anti-TIF-1? antibody-positive and negative patients, yet the sample capacity of anti-TIF-1? antibody-positive group is the smallest of 3 groups. This may generate a statistical bias if you only study the clinical characteristics of this group.
→Thank you for your comments. I understand that this data may have a bias. Therefore, we have added this bias as a limitation at the end of discussion part. Despite the small sample size, the p-value was under 0.001, as presented in table 1 and 2. I believe these data will be meaningful.
2. In your conclusion, you mentioned that clinicians should scan for cancer if patients have several manifestations. It has been known that anti-TIF-1? antibody titer is related with tumor onset, so readers may expect a more specific analysis on connections between these clinical characteristics and cancer lesion. You have only offered specific trace of one typical patient. We are expecting a more specific and statistical analysis in this part.
→Thank you for your comments. I have added the overall outcome data in Table 3. In patients who demonstrated control in cancer progression, skin lesions were stable. However, in some patients, skin lesions were controlled despite the deterioration of the primary cancer. Additionally, I added the follow-up data graph of CK. All of the patients showed decreased in level of CK. (Fig2)
Round 2
Reviewer 2 Report
please amend the word dysphasia at line 299
Author Response
Thank you for pointing it out. I apologize for this error.
